# Early Weaning Affects Liver Antioxidant Function in Piglets

**DOI:** 10.3390/ani11092679

**Published:** 2021-09-13

**Authors:** Lihuai Yu, Hongmin Li, Zhong Peng, Yuzhu Ge, Jun Liu, Tianlong Wang, Hongrong Wang, Li Dong

**Affiliations:** School of Animal and Technology, Yangzhou University, Yangzhou 225009, China; lhyu@yzu.edu.cn (L.Y.); lihongmin1124@163.com (H.L.); zweijie2021@163.com (Z.P.); xuwm321@163.com (Y.G.); lj12141018@163.com (J.L.); MZ120211426@163.com (T.W.); hrwang@yzu.edu.cn (H.W.)

**Keywords:** weaning, piglets, liver, hepatocytes, antioxidant

## Abstract

**Simple Summary:**

Early weaning is used to improve efficiency in pig production. However, early weaning may trigger liver oxidative stress in piglets. In this study, we evaluated the effects of early weaning on the development and antioxidant function of the liver in piglets. Our findings show that early weaning significantly decreases piglet body weight and suppresses liver development. We find that early weaning also suppresses the activities of superoxide dismutase (SOD) and catalase (CAT) (*p* < 0.05). It could be concluded that weaning may reduce the growth performance and liver antioxidant function of piglets.

**Abstract:**

This study examined the impact of early weaning on antioxidant function in piglets. A total of 40 Duroc × Landrace × Large White, 21-day-old piglets (half male and half female) were divided into suckling groups (SG) and weaning groups (WG). Piglets in WG were weaned at the 21st day, while the piglets in SG continued to get breastfed. Eight piglets from each group were randomly selected and slaughtered at 24th-day (SG3, WG3) and 28th-day old (SG7, WG7). The body weight, liver index, hepatocyte morphology, antioxidant enzymes activity, gene expression of antioxidant enzymes, and Nrf2 signaling in the liver of piglets were measured. The results showed that weaning caused decreased body weight (*p* < 0.01), lower liver weight (*p* < 0.01), and decreased the liver organ index (*p* < 0.05) of piglets. The area and size of hepatocytes in the WG group was smaller than that in the SG group (*p* < 0.05). We also observed that weaning reduced the activity of superoxide dismutase (SOD) and catalase (CAT) (*p* < 0.05) in the liver of piglets. Relative to the SG3 group, the gene expression of GSH-Px in liver of WG3 was significantly reduced (*p* < 0.05). The gene expression of Nrf2 in the SG3 group was higher than that in the WG3 group (*p* < 0.01). The gene expression of NQO1 in the SG7 group was higher than that in the WG7 group (*p* < 0.05). In conclusion, weaning resulted in lower weight, slowed liver development, and reduced antioxidant enzymes activity, thereby impairing liver antioxidant function and suppressing piglet growth.

## 1. Introduction

Weaning is one of the most important periods of piglet growth and may cause weaning stress. Early weaning causes physiological stress to piglets, leading to abnormal physiological and immune functions, as well as more severe oxidative stress [1,2]. The liver is the main site of piglet metabolism and contains a large number of mitochondria, where reactive oxygen species are produced [3,4]. A previous study has reported that weaning may cause liver oxidative stress in piglets, affecting liver cell apoptosis via MAPK signaling [5]. Antioxidant enzymes are important modulators of liver metabolism, and liver metabolic disorders are the main pathological factors leading to liver disease [6].

Oxidative stress is caused by reactive oxygen species (ROS), which may damage proteins, nucleic acids, and cell membranes, and affect immune responses and barrier functions in weaned piglets [7,8]. Oxidative stress affects the development and physiological functions of tissues and organs in piglets [8]. It has been reported that early weaning could reduce the activity of digestive enzymes, impair tight junctions, and damage intestinal barrier function of weaned piglets [9,10]. Weaning also disrupts the redox balance, triggering oxidative stress and apoptosis [11,12]. Some other studies have also showed that the antioxidant enzymes activity is the main indicator of the liver function in weaned piglets [13,14]. The Nrf2 signaling pathway plays an important role in the process of oxidative stress and autophagy [15,16,17]. Under homeostatic conditions, Nrf2 binds to Kelch-like ECH-associated protein 1 (Keap1) in the cytoplasm. At the same time, Keap1 binds cullin 3 and forms an E3 ubiquitin ligase complex, which can promote the occurrence of Nrf2 ubiquitination, thus causing the rapid degradation of Nrf2 [18]. In response to stress, Nrf2 will dissociate from Keap1 and then enter the nucleus and combine with the antioxidant response element (ARE), and then activate the expression of antioxidant-related genes such as NAD(P)H quinone dehydrogenase 1 (NQO1) and heme oxygenase 1 (HO-1) [19]. In this study, we examined the effects of early weaning on the expression of antioxidant-related genes and antioxidant enzymes activity in weaned piglets, which are associated with Nrf2 signaling. This study was to study the effects of early weaning on the development and antioxidant function of the liver in piglets.

## 2. Materials and Methods

### 2.1. Ethics Approval

All procedures were approved by the Institutional Animal Care and Use Committee of Yangzhou University (PR China).

### 2.2. Experimental Design

Twenty pairs of 21-day-old piglets ((Duroc × Landrace) × Large White) were chosen and randomly divided into SG and WG group. One pair includes two piglets with similar body weight from the same mother. The two piglets from the same mother were divided into two groups randomly, respectively. The sows were of similar parity (3rd or 4th) and fed with the same gestating diet that met nutrient requirements (formula were made according to the company standard). The piglets in WG group were weaned on the 21st-day old, while piglets in SG were breastfed by their own mothers until the end of the whole experiment. Eight piglets from SG and WG were randomly selected and slaughtered on the 24th-day old (SG3, WG3) and 28th-day old (SG7, WG7), respectively. The diets of the weaning piglets were designed according to the NRC (2012) (Table 1). During the experiment, the suckling group was allowed to breastfeed freely while the weaning group was given ad libitum access to the feed and water. Daily management, disinfection, and epidemic prevention were carried out according to the routine procedures of the pig farm.

### 2.3. Sample Collection

All piglets were fasted for 12 h before weighing and necropsy. The liver tissue samples were collected in 2 mL cryotubes and stored at −80 °C for later use. The formula for the calculation of organ index was: organ index (g/kg) = organ weight (g)/live weight (kg).

### 2.4. Histomorphometry Determination

Liver tissue was fixed in 4% polyoxymethylene for 48 h at 4 ℃. The tissues were then immersed in wax thrice, 40 minutes each, and embedded. They were then sectioned at 2 µm. The sections were then placed in a 37 °C water bath, dried, and stored for hematoxylin-eosin (H&E) staining. Two slices for each tissue sample were made, each slice was photographed under 10, 20, and 40× objective. A representative image of the stained part was taken on a microscope, and the size of liver cells quantified on Image-Pro Plus (6.0) (Media Cybernetics, Inc., Rockville, MD, USA). The software evaluated cell area and particle sizes of 30 cells per field of view, and finally calculated their average.

### 2.5. Determination of Activity of Antioxidant Enzymes

The study was carried out 3 and 7 days after weaning (24 and 28 days old). All piglet liver samples were collected in 2 mL cryotubes and stored at −80 °C. Liver samples (0.9–2.1 g) were homogenized in 9 volumes of normal saline on an ice water bath. They were then centrifuged at 3000 rpm for 15 min, supernatant transferred into a new centrifuge tube, and stored at 4 °C. Total oxidizing capacity (T-AOC), superoxide dismutase (SOD), catalase (CAT), glutathione peroxidase (GSH-Px), and malondialdehyde (MDA) were measured using kits that were purchased from Nanjing Jiancheng Co., Ltd. (Nanjing, China).

### 2.6. RT-qPCR Analysis

Liver samples were taken from −80 °C and total RNA extracted using TRNZOL. RNA purity (Tiangen Biotechnology, Beijing, China) was then measured using Applied Biosystems 7500 Real-Time PCR System (Applied Biosystems, Foster City, CA, USA) and RNA quality examined by agarose gel electrophoresis. RNA was then reverse transcribed into cDNA and stored it at −80 °C for later use. Extraction of total RNA and its reverse transcription were performed according to the instructions of the TAKARA kit (TAKARA RR047A, Dalian, China). Primers were designed with Primer 5.0 according to the gene sequence of the pig (http://www.ncbi.nlm.nih.gov/pubmed/ accessed on 20 December 2019). Primer sequences are shown in Table 2. The primers were synthesized by Yingwei Jieji (Shanghai, China) Trading Co., Ltd. The data was calculated using the method of 2^−ΔΔCT^ according to the reference [20].

### 2.7. Statistical Analysis

The statistical package for the Social Sciences (SPSS) 23.0 was used for statistical analysis. The body weight and liver weight of piglets were analyzed by independent t test. The other data was first analyzed by double factor variance analysis with group (G) and time (T) as fixed effects. The interaction effect of G × T was also considered. Independent t test was further used to analyze the difference between SG and WG on the different time point. Statistical significance was set at *p* < 0.05.

## 3. Results

### 3.1. Effects of Weaning on Piglet Liver Index

The results showed that the body weight, liver weight, and liver index of the WG piglets was significantly lower than that in the SG piglets (*p* < 0.05), and the liver index decreased especially on the 7th day after weaning (*p* < 0.01) (Table 3).

### 3.2. Effects of Weaning on Piglet Liver Morphology

The results showed that the particle size and area of hepatocytes in the piglets of WG group was smaller than that of the SG group (*p* < 0.05), and the area of hepatocytes decreased, especially on the 3rd day after weaning (*p* < 0.05) (Table 4).

### 3.3. Effect of Weaning on Antioxidant Enzymes Activity in Piglet Liver

The results show that the activity of SOD and CAT in the liver of WG was lower than that of SG (*p* < 0.05), and the activity of CAT decreased especially on the 7th day after weaning (*p* < 0.05) (Table 5).

### 3.4. Effect of Weaning on the Expression of Antioxidant Enzymes Genes in Piglet Liver

The results illustrate that there were no significant differences in gene expression of PRDX, SOD, and CAT between SG and WG groups (*p* > 0.05) (Table 6). However, the gene expression of GSH-Px in WG3 was significantly lower than that in SG3 (*p* < 0.05).

### 3.5. Effect of Weaning on the Expression of Nrf2 Related Genes in the Liver of Piglets

Effects of weaning on gene expression of Nrf2 pathways in the liver of piglet are shown in Table 7. The gene expression of Nrf2 in livers of WG3 was significantly lower than that in SG3 (*p* < 0.01). Compared with SG7, the expression of NQO1 in livers of WG7 was lower (*p* < 0.05).

## 4. Discussion

Weaning is an important stage in pig production. Weaning stress alters physiological and immune function of piglets, which affects piglet immune responses and intestinal barrier function [21,22]. Oxidative stress can damage mitochondrial function and increase the release of mitochondrial (ROS) [23]. The increase of ROS will increase the pro-inflammatory factors, and damage the immune system of piglets [24,25]. Previous studies have shown that weaning upsets redox balance and triggers oxidative stress in piglets [23,25]. The liver is a crucial digestive organ in piglets and contains a large number of mitochondria, which are susceptible to oxidative stress [26]. The effects of weaning on the antioxidant ability of liver are rarely searched at present. Here, we evaluated how weaning affects the growth and the antioxidant ability of liver in piglets.

The liver is an important metabolic organ that oxidizes triglycerides to produce energy and plays an important role in lipid metabolism and immune regulation [27,28]. Liver development closely correlates with the growth of pigs [29]. Pigs weaned at 28-days old exhibit better growth performance than that of pigs weaned at 21-days old [30]. The earlier weaning time will slow down the growth of piglets, which is consistent with our findings. The results in this study suggested that weaning caused the loss of body weight and liver weight on the 3rd and 7th day after weaning. The results of hematoxylin-eosin staining showed that the area and particle size of hepatocytes in WG groups was smaller than that in SG groups, indicating that weaning stress significantly affects growth of weaned piglets, slowing liver development.

Oxidative stress and reactive oxygen species ROS induce cell and tissue damage through apoptosis [31,32,33]. The liver is the main site of ROS generation. Oxidative stress influences inflammation, metabolism, proliferative liver disease, and chronic liver disease [34,35]. Enzymatic and non-enzymatic antioxidant systems are essential for cellular responses and regulate oxidative stress under physiological conditions [14,36]. Antioxidant enzymes like catalase, superoxide dismutase, and glutathione peroxidase are indicators of oxidative stress [37]. Yin et al. found that early weaning inhibited the activity of plasma SOD and CAT of piglets [38], which is consistent with our findings that the activity of SOD and CAT in liver of WG piglets is significantly lower than that of SG piglets, suggesting that weaning affects antioxidant enzymes activity in piglets.

The decrease in the expression of antioxidant mRNA in piglets is an important reason for the decreased antioxidant capacity of piglets [39]. Feng et al has reported that low-birth-weight caused decreased mRNA expression of GSH and SOD, thus caused decreased antioxidant capacity in the liver of piglets [40]. In this study, weaning down-regulated gene expression of GSH in the liver of piglets on the 3rd day after weaning. However, the enzyme activity of GSH did not change significantly after weaning, the reason for which might be that the modification and activation of antioxidant proteins are affected when piglets are under weaning stress [41,42].

Nrf2 signaling is a research hotspot in oxidative stress [43,44] and is the main regulator of cellular redox levels. Nrf2 is reported to be involved in transcriptional activation of ARE response genes, including glutathione S-transferases (GSTs), γ-glutamylcysteine synthetase (γ-GCS), heme oxygenase-1 (HO-1), antioxidants, proteasomes, and drug transporters [45,46]. Nrf2 and Keap1 form a conjugate, and then Nrf2 dissociates from the conjugate and stably translocates into the nucleus, driving expression of antioxidant enzymes genes [47,48]. In the oxidative stress model induced by diquat, Nrf2 signaling is essential for regulating the redox state of cells [49,50]. The results in this study showed that weaning caused the down-regulated expression of Nrf2 and NQO1 in the liver of piglets on the 3rd and 7th day after weaning, respectively.

## 5. Conclusions

In conclusion, early weaning reduced piglet growth, liver index, and liver antioxidant enzymes activity. Our findings indicate that weaning reduces the antioxidant capacity of liver in piglets.

## Figures and Tables

**Table 1 animals-11-02679-t001:** Composition and nutrient levels of basal diets (air-dry basis).

Ingredient	Content (%)	Nutrition Levels	Content (%)
Corn	60.50	DE (MJ/kg)	14.11
Fish meal	5.00	CP	20.21
Corn gluten meal	5.00	Ca	0.76
Soybean oil	1.00	AP	0.45
Soybean meal	24.00	Lys	1.25
Limestone	1.18	Met	0.43
CaHPO_4_	1.30	Thr	0.71
*L*-Lys	0.60	Trp	0.16
Met	0.13		
Thr	0.17		
Ser	0.02		
Choline chloride	0.10		
NaCl	0.40		
Premix ^1^	0.60		
Total	100.00		

^1^ The premix is provided for each kg of diet: VA 6000 IU, VD3 400 IU, VE 30 mg, VK3 2 mg, VB1 3.5 mg, VB2 5.5 mg, VB6 3.5 mg, VB12 25.0 μg, biotin 0.05 mg, Folic acid 0.3 mg, D-pantothenic acid 20 mg, niacin 20 mg, chloride 500 mg, iron Fe (as ferrous sulfate) 110 mg, zinc Zn (as zinc sulfate) 100 mg, Copper Cu (as copper sulfate) 20 mg, manganese Mn (as manganese sulfate) 40 mg, selenium Se (as sodium selenite) 0.30 mg, iodine I (as potassium iodide) 0.40 mg. DE Digestible energy and AP available phosphorus were calculated in reference to Nutrition parameters and Feeding Standard for Animals, while the other nutrient levels were measured values.

**Table 2 animals-11-02679-t002:** Information of Quantitative Real-time PCR.

Gene.	Accession No	Primer Sequence (5′-3′)	Amplicon Size/bp
β-Actin	DQ845171.1	AGGCCAACCGTGAGAAGATG CATGACAATGCCAGTGGTGC	122
CAT	NM_214301	GCTGAGTCCGAAGTCGTCTA CATGGTGTCCGACTAGCCTT	93
SOD	NM_214127	TGTAACTGAGCGATACGCCG GGTATTCGGCGCTCCTACAA	99
Nrf2	NM_001185152.1	TCCAAGTGAGCCATCGTTCG TGGCAGCTCATAAGGTGGTG	133
PRDX3	NM_001244531.1	CGAGACTACGGTGTGCTGTT GAGGGTCTCTTCCACGCTTC	132
HO-1	NM_001004027.1	GCCATGTGAATGCAACCCTG GCCAGTCAAGAGACCATCCC	88
Keap1	XM_021076667.1	GTGTGGAGAGGAGTCTGTGT TCCACGTTTCTGTCTCCACG	112
NQO1	NM_001159613.1	GATCATACTGGCCCACTCCG CATGGCATACAGGTCCGACA	115

**Table 3 animals-11-02679-t003:** Effects of weaning on the development of liver in piglets.

Items	Time	Groups		Mean	SEM	*p*-Value		
		S	W			G	T	G × T
Body weight (kg)	3 d	5.86 ^**^	5.08	5.47	0.18	<0.01	<0.01	0.14
	7 d	7.34 ^**^	5.77	6.56				
	Mean	6.60	5.43	6.02				
Liver weight (g)	3 d	155.43 ^*^	115.02	135.23	5.30	<0.01	0.01	0.11
	7 d	195.63 ^**^	129.95	162.79				
	Mean	175.53	122.49	149.01				
Liver index (g/kg)	3 d	26.47	22.79	24.63	0.18	<0.01	<0.01	0.14
	7 d	27.12 ^*^	22.52	24.82				
	Mean	26.80	22.66	24.73				

Values are expressed as means and SEM, n = 8; S, suckling group; W, weaning group; G, group; T, time. **p* < 0.05, ***p* < 0.01 (* Indicates significant differences between different treatments at the same weaning time).

**Table 4 animals-11-02679-t004:** Effects of weaning on the size and area of liver cells of piglets.

Items	Time	Groups		Mean	SEM	*p*-Value		
		S	W			G	T	G × T
Particle size (μm)	3 d	6.51 ^*^	4.26	5.39	0.93	0.04	0.08	0.88
	7 d	8.02 ^*^	6.06	7.04				
	Mean	7.27	5.16	6.22				
area (μm^2^)	3 d	87.85 ^*^	80.6	84.23	1.73	0.01	0.06	0.72
	7 d	91.02	86.06	88.54				
	Mean	89.44	83.33	86.39				

Values are expressed as means and SEM, n = 8; S, suckling group; W, weaning group; G, group; T, time. * *p* < 0.05 (* Indicates significant differences between different treatments at the same weaning time).

**Table 5 animals-11-02679-t005:** Effects of weaning on antioxidant function of liver of piglets.

Items	Time	Groups		Mean	SEM	*p*-Value		
		S	W			G	T	G × T
T-AOC (U/mg prot)	3 d	3.52	3.29	3.41	0.09	0.13	0.84	0.73
	7 d	3.62	3.27	3.45				
	Mean	3.57	3.28	3.43				
SOD (U/mg prot)	3 d	5.80 ^*^	5.63	5.72	0.02	<0.01	0.73	0.76
	7 d	5.80 ^*^	5.60	5.70				
	Mean	5.80	5.62	5.71				
CAT (U/mg prot)	3 d	12.92	11.64	12.28	0.33	0.02	0.65	0.50
	7 d	13.06 ^*^	10.89	11.98				
	Mean	12.99	11.27	12.13				
GSH-Px (U/mg prot)	3 d	209.46	188.83	199.15	9.26	0.10	0.12	0.55
	7 d	191.09	147.68	169.39				
	Mean	200.28	168.26	184.27				
MDA (nmol/mg prot)	3 d	1.83	2.29	2.06	0.13	0.11	0.31	0.87
	7 d	1.61	1.99	1.80				
	Mean	1.72	2.14	1.93				

Values are expressed as means and SEM, n = 8; S, suckling group; W, weaning group; G, group; T, time. * *p* < 0.05 (* Indicates significant differences between different treatments at the same weaning time).

**Table 6 animals-11-02679-t006:** Effects of weaning on the gene expression of antioxidant enzymes in the liver of piglets.

Items	Time	Groups		Mean	SEM	*p*-Value		
		S	W			G	T	G × T
PRDX	3 d	1.00	1.02	1.01	0.09	0.71	0.04	0.95
	7 d	1.46	1.41	1.44				
	Mean	1.23	1.22	1.22				
SOD	3 d	1.00	0.83	0.92	0.39	0.46	0.04	0.60
	7 d	3.10	2.08	1.70				
	Mean	2.05	1.46	2.59				
CAT	3 d	1.00	0.89	0.95	0.24	0.78	0.12	0.92
	7 d	1.79	1.70	1.75				
	Mean	1.40	1.30	1.35				
GSH-Px	3 d	1.00 ^*^	0.44	0.72	0.22	0.57	<0.01	0.42
	7 d	2.43	2.54	2.49				
	Mean	1.72	1.49	1.61				

Values are expressed as means and SEM, n = 8; S, suckling group; W, weaning group; G, group; T, time. **p* < 0.05 (* Indicates significant differences between different treatments at the same weaning time).

**Table 7 animals-11-02679-t007:** Effects of weaning on gene expression of Nrf2 pathways in the liver of piglet.

Items	Time	Groups		Mean	SEM	*p*-Value		
		S	W			G	T	G × T
Nrf2	3 d	1.00 ^**^	0.31	0.66	0.66	0.71	0.07	0.87
	7 d	3.31	3.04	3.18				
	Mean	2.16	1.68	1.92				
HO-1	3 d	1.00	1.08	1.04	0.54	0.93	0.19	0.27
	7 d	1.17	0.86	1.02				
	Mean	1.09	0.97	1.03				
NQO1	3 d	1.00	0.87	0.94	0.52	0.65	0.07	0.06
	7 d	2.40 ^*^	0.16	1.28				
	Mean	1.70	0.52	1.11				
Keap1	3 d	1.00	1.21	1.11	0.51	0.05	0.19	0.78
	7 d	1.36	1.69	1.53				
	Mean	1.18	1.45	1.32				

Values are expressed as means and SEM, n = 8; S, suckling group; W, weaning group; G, group; T, time. **p*< 0.05, ***p*< 0.01 (* Indicates significant differences between different treatments at the same weaning time).

## Data Availability

The data sets analyzed in the present study are available from the corresponding author on reasonable request.

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
