# Peer review of "Early Weaning Affects Liver Antioxidant Function in Piglets"

_animals, 2021, doi:10.3390/ani11092679_

Round 1

Reviewer 1 Report

Dear authors,

thank you for your work.

I have ranked your work into "major revison". However, in general, I have some doubts that the power of your project is sufficient in order to analyze differences between expression levels.

Details:

L53: A more detailed description of the Nrf2-pathway would be useful.

L60: “Large” Large White?

L62: Were the males castrated or not?

L62: Are the litter sizes of the sow available? This would be a useful information.

L65: Do the 8 selected piglets come from different sows or litters? Please provide this information.

L74: Units of "nutrition levels" are missing.

L80: „The nutrition level is calculated value.“ What does that mean?

L89: Typo: ..

L92: „Roundness“: More explanation about this trait is needed

L114: I assume that birth weight of the piglet has an important influence on many traits of your study. If this information is available, I would be (very) useful to include BW as a co-variable into the statistical model.

L114: Did you check if the ND (Normal Distribution) assumption fits?

L155: What is the unit of the gene expression? How did you analyze this data? (Delta Delta Ct method?) More explanation needed?

L160: You have stated, that the difference between S and W group in Nrf2 expression is significant, but the main factor G is not. How can that be? In general, if this factor is non-significant, it is not allowed to interpret differences between sub groups.

L168: Details which describes the relationship between "immune function" and "oxidative stress" can be found in the literature. Please provide at least some details of this aspect.

L177: „Piglets weaned at 14 days of age have significantly higher body weight than those weaned at 7 days of age“. This is a trivial finding. May be I have msisunderstood.

„… and that weaning before 14 days of age significantly reduces piglet growth and affects their wellbeing, which is consistent with our findings“ You did not analyses this aspect in your study

L204-206: The group effect as well as the interaction G x T was non significant. So it is not allowed to interpret this difference.

Author Response

Dear Reviewer,

Thank you very much for your kind comments from yourself.

We have carefully considered the comments of you and have revised our manuscript according to the advice that we were given.Please see the attachment.

If you have any more questions about this paper, please contact us without hesitation. Thank you very much and all the best to you and your family.

Sincerely yours,

Hongmin Li

Reviewer 2 Report

This manuscript explores how weaning impacts liver morphology and function in young pigs. Although this manuscript has an interesting topic, I have several fundamental flaws with this manuscript that must be remedied in order for it to meet publishing standards. Firstly, the entire crux of the manuscript is based on differences in Nrf2 signaling. However, there is simply not the data to back this conclusion up. Based on the tables, there is no treatment effect of group on Nrf2 abundance, except for perhaps at 3d – however this is based off somewhat vague statistical analysis. Further, there is no difference in weaning groups for any of the other Nrf2 related genes. Thus, the authors need to reframe the manuscript to better describe their actual findings. Further, the authors make several false claims regarding their NQO1 abundance findings and refer several times to NAD(P)H abundance, which was not presented. This must be fixed throughout.

My second big issue with this manuscript is how results were analyzed and presented. It appears the authors analyzed the data as a 2 x 2 factorial to look at the effects of group, time, and their interaction. However, from there it gets a bit fuzzy. It looks like the authors performed individual contrasts among treatment groups, however they do not do a very good job of explaining this. Further, this data would be better presented based on complete means separation from within the ANOVA, as it would allow for adjustments to be made for multiple comparisons. It is not valid to compare all the treatment groups without adjusting for this. Further, the authors do not do a good job of presenting their results in the text or in tables. Thus, significant overhaul should be made to the statistical analysis and results section.

Specific comments are below:

Title: Remove Nrf2 signaling information.

L13: you did not examine NAD(P)H gene abundance and NQO1 abundance was not affected by weaning.

L14: reduce

L22 and throughout: please include specific P-values if referring to only 1 parameter. Additionally, please change =< to ≤

L26-27: again, this is false

L29: “abnormal development”

L36-37: these references (1 and 2) do not include anything about immune status or oxidative stress. Please remove these statements or find references that do in fact discuss immune status and oxidative stress.

L38-39: reactive oxygen species do not “function”. Perhaps “are generated” or “are produced” would be better statements.

L47: digestive enzymes

L60: should this be “x Large White”?

L64: weaned at 21 days

L68: “Nutritional Requirements of Swine (NRC, 2012) piglet nutritional requirements and production practice”. What do the authors mean by “production practice”? please clarify or remove.

L69: Why were weaned piglets only fed 3x/day rather than given ad libitum feed access? How can you ensure that weaned piglets would not consume more if given ad libitum access?

L81: Were piglets fasted prior to necropsy? This information is relevant, as liver status can change quickly in response to feed events and is particularly important given that suckling pigs had ad libitum access to nutrition whereas weaned pigs were restrict-fed.

L87: add period after first sentence

L89: remove a period after “staining”

L89-93: change the order of some of these sentences so the methods make more sense. “making tissues slices” and “photographing images” should not come before analysis and quantification

Statistics and Results: The authors and tables do not do a very good job of demonstrating statistical significance with respect to the different tests performed and presenting their results. Thus, this section is confusing and messy. Based on the statistical analysis section, it appears the authors utilized a 2 x 2 factorial design to examine the effects of weaning, time, and their interaction. However, they also claim to have performed a separate, independent test looking at the main effect of weaning. Was this test performed looking at the two timepoints together, or at individual timepoints? If individual timepoints were examined, then individual contrasts should have been constructed or the authors present this data as a factorial analysis, and differentiating treatment groups by complete means separation within this analysis. Regardless, comparing treatment groups individually without adjusting for multiple comparisons is not a valid approach.

Additionally, the tables do not do a very good job of demonstrating which means are different. Do the asterisks denote significant differences between weaning groups at specific (d3 or d7) timepoints based on the independent statistical test? If so, this needs to be made more clear so that the reader can determine for themselves which groups are significantly different.

Finally, the authors do a very poor job of explaining their results. It is very unclear whether or not the authors are talking about main effects of weaning, regardless of time, or if they are talking about the effects of weaning at specific timepoints. The authors need to carefully go through and reword the results section to make this more clear.

L123: The value of piglet weight depends on whether or not pigs were in the same nutritional state when necropsied (see above)

L123-126: These 4 sentences say basically the same thing 4 different times. This is unnecessary. Further, there is no discussion of time effects, or the interactions of weaning and time. Please re-write to avoid the redundancies while also incorporating time effects into your results section and/or discussing the effects at specific timepoints.

L131: “Hepatocyte area and size in the WG3 group was smaller than in the WG7 group (p=<0.05)” based on your statistical analysis, you did not perform complete means separation NOR did you examine the effects of time within treatment. Thus, you cannot include this statement in your results.

L141-145: if activity assays were performed, then “levels” is not an adequate descriptor.

L142-145: again, clarify to eliminate redundancy or make more clear

Table 6: there is no effect of treatment for GSH-Px activity, so where is this significance asterisk coming from? Again, if the authors had performed complete means separation, this would be more clear.

L160: based on your P-values, there is no Group significance when it comes to Nrf2 abundance. Thus, your statement in line 160 is inaccurate.

L161: lesser in WG3 pigs compared to which SG pigs? The average of SG3 and SG7, just SG3?

L162: this statement is false. the time effect for NQO1 abundance was not significant. Rephrase.

L174: you did not measure liver oxidative stress; no ROS and/or oxidative damage indicators were evaluated. Rephrase.

L178-179: Use a better reference. There are many studies that examine weaning age and its effect on growth at ages more relevant to the current study (18-25 days of age), 14 vs 7 days of age for weaning is not the same as what was done in the current study and isn’t necessarily industry relevant either.

L180-181: “Our data show that weight and liver coefficient of piglets weaned on the 3rd and 7th day were lower than those of normal SG groups” this statement does not represent your dataset very well. Pigs weren’t weaned on two different dates, they were weaned at d21 and evaluated 3 and 7 days post-weaning. Rephrase to more accurately reflect this.

L204-205: “We find that the expression of Nrf2 and NQO1in WG piglets was significantly lower than in SG piglets” this is not consistent with your

L185-194: you measured activity, not content. Please alter your wording to reflect this.

L203-204: how does PRDX relate to your discussion of Nrf2? Connect these ideas or remove this line.

L205-208: GSH mRNA abundance differed (sort of), yet activity was unaltered. This discrepancy should be discussed.

L209-212: reduced antioxidant activity is not indicative of increased oxidative stress, please remove this statement. Further, remove the conclusions regarding Nrf2.

L213: conclusions. Do you intend to have a conclusions section or will you include that into the discussion? Please either add a conclusions section or delete this from the manuscript

L216-222: please include COI, funding, author contributions, data availability, etc.

Author Response

(The authors gave the same response as above.)

Round 2

Reviewer 2 Report

L54: change “thus caused the rapid…” to thus causing the rapid degradation…”

L69: please remove the “and” from the beginning of the sentence

L70: “birth order” – do you mean parity?

L77-79: “while the weaning group was given ad libitum access, was given ad libitum access, with ad libitum access to drinking water.” – please rephrase this, it doesn’t make sense.

Table 2: why did you add a comma and “m” to your primer sequence?

L154: change to “..there were no significant differences in gene expression…”

L162: change to “…expression of Nrf2…”

L181: change 28-day old to 28-days old and 21-day old to 21-days old

L186: change hepatocyte to hepatocytes

Author Response

Dear Reviewer,

Thank you very much for your kind comments from yourself.

We have carefully considered the comments of you and have revised our manuscript according to the advice that we were given.Please see the attachment.

If you have any more questions about this paper, please contact us without hesitation. Thank you very much and all the best to you and your family.

Sincerely yours,

Hongmin Li

Response to Reviewer 2

Comments - 1) L54: change “thus caused the rapid…” to thus causing the rapid degradation…”

Response – Thank you for your suggestion. We have changed “thus caused the rapid…” to thus causing the rapid degradation…”. The revision sentence is “thus causing the rapid degradation of Nrf2.”

Comments - 2) L69: please remove the “and” from the beginning of the sentence

Response – Thank you for your suggestion. We have deleted “and” from the sentence. The revision is bellow.

“The two piglets from the same mother were randomly divided into two groups, respectively.”

Comments - 3) L70: “birth order” – do you mean parity?

Response – Thank you for your suggestion. Yes, you are right. We have replaced “birth order” with “parity”. The revision could be seen as follows.

“The sows were of similar parity (3rd or 4th) and fed with the same gestating diet that met nutrient requirements (formula were made according to the company standard).”

Comments - 4) L77-79: “while the weaning group was given ad libitum access, was given ad libitum access, with ad libitum access to drinking water.” – please rephrase this, it doesn’t make sense.

Response – Thank you for your suggestion. We have rephrased this sentence. The revision sentence is “while the weaning group was given ad libitum access to the feed and water.”

Comments - 5) Table 2: why did you add a comma and “m” to your primer sequence?

Response – Thank you for your suggestion. Sorry for the mistake we have deleted the comma and “m”.

Comments - 6) L154: change to “..there were no significant differences in gene expression…”

Response – Thank you for your suggestion. We have made revision on this sentence. The revision could be seen as follows.

“The results illustrate that there were no significant differences in gene expression of PRDX, SOD and CAT between SG and WG groups.”

Comments - 7) L162: change to “…expression of Nrf2…”

Response – Thank you for your suggestion. We have replaced “gene expression Nrf2” with “gene expression of Nrf2”.

Comments - 8) L181: change 28-day old to 28-days old and 21-day old to 21-days old

Response – Thank you for your suggestion. We have made revision on this sentence. The revision could be seen as follows.

“Pigs weaned at 28-days old exhibit better growth performance than that of pigs weaned at 21-days old.”

Comments - 9) L186: change hepatocyte to hepatocytes

Response – Thank you for your suggestion. Yes. We have replaced “hepatocyte” with “hepatocytes”.